# ENTROPY-GUIDED AUTOMATED PROGRESSIVE PRUNING FOR DIFFUSION AND FLOW MODELS

## ABSTRACT

Large-scale vision generative models, including diffusion and flow models, have demonstrated remarkable performance in visual generation tasks. However, transferring these pre-trained models to downstream tasks often results in significant parameter redundancy. In this paper, we propose EntPruner, an entropy-guided automatic progressive pruning framework for diffusion and flow models. First, we introduce entropy-guided pruning, a block-level importance assessment strategy tailored for transformer-based diffusion and flow models. As the importance of each module can vary significantly across downstream tasks, EntPruner prioritizes pruning of less important blocks using data-dependent transfer entropy as a guiding metric. Second, leveraging the entropy ranking, we propose a zero-shot Neural Architecture Search (NAS) framework during training to automatically determine when and how much to prune. This dynamic strategy avoids the pitfalls of one-shot pruning, mitigating mode collapse, and preserving model performance. Extensive experiments on DiT and SiT models demonstrate the effectiveness of EntPruner, achieving up to 2.22× inference speedup while maintaining competitive generation quality on ImageNet and three downstream datasets.

## 1 INTRODUCTION

A myriad of recent breakthroughs in diffusion and flow models have demonstrated their remarkable capabilities in image generation. The success has also been extended to audio, video, and language domains (Huang et al., 2023; Zhu et al., 2024; Sahoo et al., 2024). The Denoising Diffusion Probabilistic Model (DDPM) (Ho et al., 2020) highlighted the effectiveness of the U-Net backbone. Recently, owing to the superior performance of Diffusion Transformers (Peebles & Xie, 2023), studies have increasingly adopted transformer-based architectures. Lipman et al. (Lipman et al., 2022) further introduced Flow Matching, a more direct and faster generative trajectory that offers an alternative perspective on training and inference for diffusion models. While diffusion models have evolved rapidly in recent years, achieving near-photorealistic quality, their practical deployment remains limited due to computational inefficiency. Recent trends in architectural design—particularly the adoption of transformer-based backbones like DiT (Peebles & Xie, 2023) and SiT (Ma et al., 2024)—have significantly improved scalability and expressivity. However, these advancements come at the cost of increased parameter counts and memory usage. These models suffer from efficiency issues when deployed on edge devices or used in low-latency settings such as interactive applications. The high computational cost and slow inference speed of diffusion and flow models motivate us to explore more effective solutions.

To solve these problems, a majority of the arts have explored efficient pruning strategies for Stable Diffusion (SD) models (Rombach et al., 2022). BK-SDM (Kim et al., 2024) employs a heuristically handcrafted pruning scheme and leverages distillation to recover performance after pruning. However, its manual design limits transferability and requires substantial human efforts and computational costs. Diff-Pruning (Fang et al., 2023) removes filters by identifying unimportant weights through gradient analysis, but its threshold must be tuned for specific tasks, restricting practical applicability. LD-Pruner (Castells et al., 2024) introduces a novel metric to evaluate the importance of each operator and prunes unimportant convolutional and attention layers, but this metric is task-independent. Furthermore, none of these methods have been examined on transformer-based diffusion models, leaving their effectiveness uncertain.

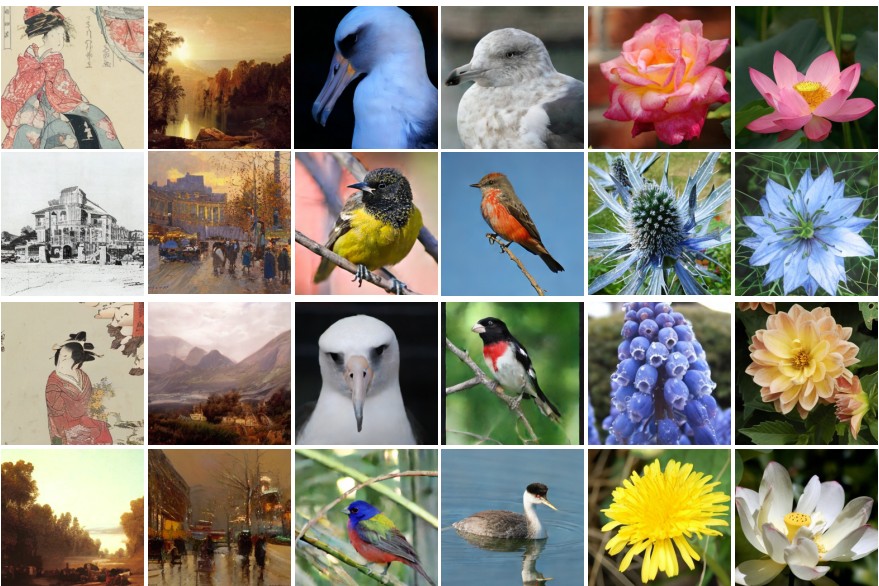

Figure 1: **Generated results of a series of *flow models* pruned by EntPruner.** Base model is SiT-XL/2. The pruning rates in rows 1 and 2 are set to 35%, while those in rows 3 and 4 are set to 50%. Inference is performed on the ArtBench (column 1-2), CUB (columns 3-4), and Flowers (columns 5-6) datasets, with the classifier-free guidance coefficient set to 4.0. The sampling process involves 250 steps, and ODE solver is used.

Training a Large DiT from scratch is prohibitively costly—for example, generating 256×256 ImageNet images requires roughly 4,700 GPU-hours (Yao et al., 2024). Fine-tuning is the main approach for transfer learning tasks. Although various fine-tuning techniques have been proposed to accelerate the training process, they typically retain the computational cost of the original model during inference (Xie et al., 2023; Li et al., 2024a). Our observations suggest that transferring large pretrained models to downstream tasks often introduces substantial parameter redundancy, as smaller or less complex tasks rarely require the full capacity of the original model. Moreover, many downstream tasks—such as domain-specific generation or low-resolution synthesis—do not benefit from the full capacity of large-scale pretrained models.

These discrepancies raise a critical challenge: how to adaptively compress diffusion models without compromising their generative quality? Current pruning methods fail to address this in a task-aware and scalable manner, especially when applied to transformer-based diffusion backbones. Furthermore, prior pruning work has also demonstrated a delicate trade-off between model performance and pruning ratio: aggressive pruning often leads to instability and catastrophic forgetting, making the post-pruning training process slow and ineffective. These challenges highlight the need for a principled, efficient, and robust pruning framework that can adapt to various downstream settings.

Numerous studies have demonstrated that not all parameters in deep neural networks contribute equally to model performance (Cheng et al., 2023; Guo et al., 2020; Yang et al.). As shown in Figure 2, we evaluate the contribution of each block in a pre-trained SiT-XL/2 model (trained on ImageNet) to downstream task performance. Transfer entropy is particularly well-suited for evaluating information flow and inter-layer dependency in sequential generation models like diffusion transformers. Unlike simple magnitude-based or gradient-based importance measures, transfer entropy captures the directional influence of each block on the output distribution, thereby offering a more nuanced understanding of a block's contribution to the model's expressive power. Figure 2 (a) and (b) respectively report the change in training loss and transfer entropy when individual blocks are removed. The results indicate significant variation in how different blocks contribute to the model's expressivity and convergence speed. Figure 2 (c) and (d) further reveal a strong positive correlation between loss and transfer entropy. Moreover, as the number of removed blocks increases, both loss and transfer entropy tend to rise. The large variance observed across random removal trials suggests that the impact of indiscriminate pruning is highly unpredictable and often detrimental to convergence, making the model harder to train effectively.

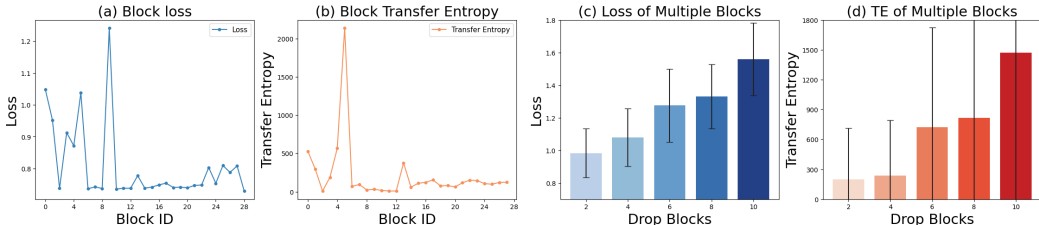

Figure 2: **Block Interactions in SiT.** We apply masking to each block of the pre-trained SiT-XL/2 model individually and compute the corresponding transfer entropy and output loss, as shown in (a) and (b). Blocks with higher influence—indicated by higher transfer entropy—contribute more to faster convergence and better model performance. In (c) and (d), we randomly mask blocks in the pre-trained SiT-XL/2 model 20 times and compute the mean and variance of the resulting loss. A strong positive correlation is observed between transfer entropy and the associated loss, highlighting the effectiveness of transfer entropy in quantifying block importance.

We propose a task-aware pruning framework for Transformer-based diffusion models, where each block primarily consists of MLP and self-attention layers. The framework performs block-level structured pruning through a two-stage process: **1)** Entropy-Guided Importance Estimation: We evaluate block interactions in the pretrained diffusion model using transfer entropy, which enables us to rank blocks based on their contribution to the overall model behavior. **2)** Automatic progressive pruning: We first introduce a linear pruning scheme, based on which we introduce an erasure operator to determine when and how much to prune by solving the optimization problem. To further reduce computational overhead, we integrate multiple zero-shot Neural Architecture Search (NAS) metrics, allowing us to reformulate pruning as a sub-network architecture search problem that can be solved with minimal cost.

Our framework accurately identifies the least important components for each downstream task, thereby minimizing disruption to the pretrained weights. It autonomously determines the optimal pruning schedule at different training stages, resulting in faster convergence while maintaining near-lossless performance. Through extensive experiments on multiple benchmark datasets and two widely-used diffusion architectures (DiT and SiT), we show that our entropy-guided pruning framework not only reduces model size and inference cost but also achieves superior or comparable generative performance compared to full fine-tuning and existing pruning baselines (Figure 1). This work bridges a gap in the compression of diffusion Transformers and offers a practical solution for their deployment in resource-constrained environments.

The main contributions are summarized as follows:

- We introduce **Transfer Entropy (TE)** to evaluate the interaction between blocks and rank the importance of each block, so that the network performance will be less or even not degraded during the pruning process.

- We further propose an **Entropy-Guided Automatic Pruning Framework (EntPruner)**, which can automatically select the optimal pruning time and pruning rate during training, making the post-pruning training process more stable and less loss of performance.

- Our method is validated on a wide range of benchmark datasets, with an average FID drop of only 1.76 at a 50% pruning rate, while the inference speed is **2.22**$\times$ faster than the full model. These excellent experimental performances demonstrate the potential of our method for different diffusion models and downstream tasks.

## 2 ENTROPY-GUIDED PROGRESSIVE PRUNING FOR DIFFUSION AND FLOW MODELS

### 2.1 DIFFUSION AND FLOW MODELS

Both diffusion and flow models corrupt data $x_* \sim p(x)$ by progressively injecting noise $\epsilon \sim \mathcal{N}(0, I)$. A unified forward process can be written as $x_t = \alpha_t x_* + \sigma_t \epsilon$, where $\alpha_t$ and $\sigma_t$ are time-dependent functions. In score-matching diffusion models, the process is usually defined on

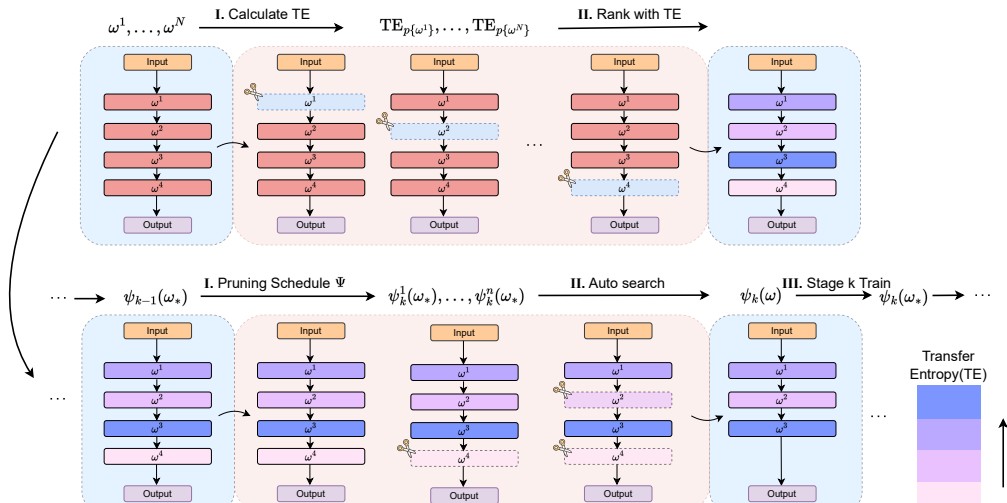

Figure 3: **Entropy-guided automatic pruning framework.** At first, we employ transfer entropy to evaluate and rank the expressiveness of individual blocks, where darker regions in the heatmap indicate stronger interactions with the overall network. At each pruning stage, the pruning schedule explores candidate subnetworks under different pruning ratios, selects the optimal one using zero-shot NAS proxies, and inherits parameters from the previous stage.

discrete time steps, and $x_t$ converges to $\mathcal{N}(0, I)$ as $t \to \infty$. Flow-matching methods, in contrast, restrict the trajectory to $t \in [0, 1]$ and set $\alpha_0 = \sigma_1 = 1$ and $\alpha_1 = \sigma_0 = 0$, so that $x_t$ converges to $\mathcal{N}(0, I)$ precisely at $t = 1$. In the reverse process, both flow matching and score-based diffusion models can be implemented by learning to invert the forward dynamics using either a stochastic differential equation (SDE) (Albergo & Vanden-Eijnden, 2022) or probability flow ordinary differential equation (ODE) (Albergo et al., 2023).

## 2.2 OVERVIEW.

Our goal is to enhance the efficiency of Transformer-based diffusion models (DMs) by removing blocks with minimal interactions, thereby achieving model lightweighting without sacrificing performance. To this end, we first need to establish a pruning priority: given a target pruning ratio $r$, less informative blocks should be pruned first, while more critical blocks are preserved. Transfer entropy (TE) (Schreiber, 2000; Lin et al., 2024b) provides a reliable measure of both the information contained in each block and its complex interactions within the network. By leveraging transfer entropy, we can quantitatively assess block importance and systematically determine which less essential blocks should be pruned. Prior studies have shown that a large pruning rate often disrupts the pretrained model's prior, leading to irreversible performance degradation. Conversely, an appropriate pruning ratio can effectively balance efficiency and accuracy. Therefore, we propose a progressive pruning framework. We first define a pruning schedule $\Psi$ for progressive pruning, which consists of a sequence of sub-networks, and the pruning space set is $\xi$. Let $\mathcal{L}$ denote the task loss, $\Omega$ the model size, and $\omega$ the set of model parameters. The objective of progressive pruning can be formulated as:

$$\min_{\omega, \Psi, \Gamma} \{\mathcal{L}(\omega, \Psi), \, \Omega\}. \tag{1}$$

To reduce the complexity of optimizing over $\Psi$, we adopt a linear pruning space based on a predefined pruning ratio. To ensure that the model is sufficiently optimized after each pruning step, the whole pruning process is divided into $k$ stages (default $k = 4$), with each stage consisting of $s = S/k$ training iterations. For a target pruning ratio $r$, the schedule is defined as $\Psi = \{\psi_i\}_{i=1}^{|k|}$, where $\psi_k$ retains $(1 - r)\%$ of the original parameter count. Throughout training, the model size is gradually reduced according to $\Psi$.

Pruning redundant operators in predefined neural networks to search for optimal sub-networks is a key research direction in the field of Neural Architecture Search (NAS) (Chen et al., 2023b). Exsisting studies have investigated how to identify important sub-networks starting from a supernet

that encompasses all candidate connections and operations (Wang et al., 2023; Li et al., 2023a). We optimize the pruning schedule $\Psi$ via zero-shot NAS, where each candidate subnetwork is evaluated by NTK condition numbers and ZiCo scores. These metrics jointly capture convergence properties and gradient stability, enabling $\Psi$ to automatically select and prune less informative blocks while preserving the trainability of the subnetwork.

## 2.3 Quantifying the Importance of Blocks with Transfer Entropy

**Entropy Quantification.** Given a denoising network, let the distribution of output feature maps be denoted as $p(x)$, $x \in \mathbf{X}$, where $\mathbf{X}$ represents the set of output feature maps from the network. The entropy can be expressed as:

$$H(\mathbf{X}) = -\int p(x) \log p(x) \, dx, \quad x \in \mathbf{X}. \tag{2}$$

For tractability, we assume that $p(x)$ follows a Gaussian distribution, i.e., $\mathbf{X} \sim \mathcal{N}(\mu, \sigma^2)$. Eq.( 2) can be written as:

$$\begin{aligned}
H(\mathbf{X}) &= -\mathbb{E}[\log \mathcal{N}(\mu, \sigma^2)] \\
&= -\mathbb{E}[\log[(2\pi\sigma^2)^{-1/2} \exp(-\frac{1}{2\sigma^2}(f - \mu)^2)]] \\
&= \log(\sigma) + \frac{1}{2}\log(2\pi) + \frac{1}{2}
\end{aligned} \tag{3}$$

**Interaction among Blocks.** The complexity of interactions between layers with diffusion models can serve as an indicator of their importance to the whole network. However, this cannot be implemented by simple entropy quantification. Transfer entropy(TE) (Schreiber, 2000; Lin et al., 2024b) addresses this limitation by quantifying the information discrepancy between a original and a target network. The formulation of TE is defined as:

$$\begin{aligned}
\text{TE} &= H(\mathbf{X}_{\text{original}}) - H(\mathbf{X}_{\text{target}}) \\
&= H(\mathbf{X}_{\text{out}}) - H(\mathbf{X}_{\text{out}} \mid \text{Mask}\{block_i\}),
\end{aligned} \tag{4}$$

where $H(\mathbf{X}_{\text{out}})$ denotes the entropy of the original network output, and $H(\mathbf{X}_{\text{out}} \mid \text{Mask}\{block_i\})$ represents the entropy after masking the $i^{th}$ block. Therefore, Eq. (4) provides a metric to evaluate the effect of removing a certain block on the entire pretrained model.

## 2.4 Automatic Pruning via Zero-Shot Neural Network Search.

Assume that $\mathcal{H}$ is a zero-shot performance predictor used to estimate the loss of each candidate sub-network. The optimal sub-network at stage $k$ can be identified by solving:

$$\begin{aligned}
\psi_k^* &= \underset{\psi_k \in \Lambda_k}{\arg\min} \left\{ \mathcal{H}(\omega(\psi_k)), \ \Omega \right\}, \\
\text{where} \quad \Lambda_k &= \{\psi \in \xi \mid |\omega(\psi)| \leq |\omega(\psi_{k-1})|\}.
\end{aligned} \tag{5}$$

where $\Lambda_k$ denotes the set of candidate sub-networks whose parameter sizes are no larger than that of $\psi_{k-1}$, the sub-network from the previous pruning stage.

**Trainability via the NTK Condition Number in Flow Matching.** The trainability of a neural network reflects how effectively it can be optimized via gradient descent. While larger models offer more expressivity, this does not guarantee practical trainability. The Neural Tangent Kernel (NTK) provides a useful tool for analyzing convergence in the infinite-width regime (Jacot et al., 2018; Novak et al., 2022).

In flow matching, a denoising network learns the velocity field $v(t, x_t)$ that transports noisy samples $x_t = \alpha_t x_* + \sigma_t \epsilon$ toward data $x_*$. The true velocity is $v(t, x_t) = \dot{\alpha}_t x_* + \dot{\sigma}_t \epsilon$, and the model $v_\theta(x_t, t)$ is trained to approximate it. For a candidate sub-network with parameters $\omega$, the update of predicted velocity satisfies

$$\Delta v(x_t) = -\eta \hat{\Theta}(x_t, x_t) \nabla v(x_t) \mathcal{L}, \tag{6}$$

where $\hat{\Theta}$ is the NTK of the velocity predictor.

In the infinite-width limit, the training dynamics are governed by the eigenvalues $\{\lambda_i\}$ of $\hat{\Theta}$. The maximum stable learning rate scales as $\eta \sim 2/\lambda_0$, while the convergence of the slowest mode depends on $1/\kappa$, where $\kappa = \lambda_0/\lambda_m$. A smaller $\kappa$ indicates faster and more stable convergence. Thus, we adopt the NTK condition number as a zero-shot NAS metric:

$$\mathcal{H}_\kappa(\psi) = \frac{\lambda_0}{\lambda_m}. \tag{7}$$

where $\lambda_0$ and $\lambda_m$ denote the largest and smallest eigenvalues of the NTK, respectively. More details are provided in Appendix A.3.

**Convergence Rate and Generalization Capacity via Gradient Analysis.** Convergence and expressivity also directly influence the final performance of a neural network. After a number of training steps, a network with a larger absolute mean of gradient and smaller standard deviation of gradient is generally associated with lower training loss and faster convergence. Interestingly, a smaller gradient variance often correlates with a lower maximum eigenvalue $\Theta$ of the NTK, which implies a smoother loss landscape and better generalization performance (Lewkowycz et al., 2020).

We adopt ZiCo as one of the zero-shot NAS metrics, which jointly considers the absolute mean and standard deviation of gradient. The parameters of a candidate sub-network $\omega$ are also inherited from the trained parameters $\omega_*$ of the previous pruning stage. Since the ZiCo metric has been shown to correlate positively with network trainability, we introduce a negative sign to make it positively correlated with loss, allowing it to be minimized during optimization:

$$\mathcal{H}_{\text{ZiCo}}(\psi) = -\sum_{l=1}^{N} \log \left( \sum_{\omega \in \omega_l} \frac{\mathbb{E}\left[|\nabla_\omega \mathcal{L}^*|\right]}{\sigma\left(|\nabla_\omega \mathcal{L}^*|\right)} \right). \tag{8}$$

where $N$ denotes the number of layers in the candidate sub-network, $\omega_l$ is the set of parameters in the $l^{\text{th}}$ layer, $\mathcal{L}^*$ is the loss $\mathcal{L}(\mathbf{x}_{t,i}, \mathbf{v}_{t,i}; \omega)$, $i \in \{1, \ldots, D\}$ and $D$ is the number of training batches, typically set to 2 to balance stability and efficiency.

## 2.5 ENTROPY-GUIDED AUTOMATIC PROGRESSIVE PRUNING FRAMEWORK

We summarize the overall algorithmic workflow of EntPruner as follows. First, we perform block-level importance ranking using transfer entropy to evaluate the contribution of each block in the pretrained model. Second, we employ two zero-shot NAS proxies to guide the pruning schedule at each training stage. The optimization objective, as reformulated from Eq. (5) is:

$$\psi_k^\star = \arg\min_{\psi_k \in \Lambda_k} \left\{ \mathcal{H}_\kappa(\psi_k), \ \mathcal{H}_{\text{ZiCo}}(\psi_k), \ \Omega \right\}. \tag{9}$$

A key challenge lies in how to jointly optimize the two proxies. We assume both proxies are equally important for maintaining network performance and training efficiency. A common strategy is to apply a voting-based algorithm to select the optimal candidate sub-network. This approach helps mitigate differences in scale between the two metrics. Additionally, we incorporate model' parameters as a regularization term in the selection process. The final optimization is defined as:

$$\psi_k^\star = \arg\min_{\psi_k \in \Lambda_k} R(\psi_k),$$
$$\text{s.t.} R(\psi_k) = R(\mathcal{H}_\kappa(\psi_k)) + R(\mathcal{H}_{\text{ZiCo}}(\psi_k)) + \gamma R(\Omega). \tag{10}$$

where $R(\cdot)$ denotes the ranking score (*e.g.*, 1st, 2nd, ..., $R$-th) and $\gamma$ is the regularization factor default set to 0.5. The candidate with the lowest rank in each individual metric receives the smallest score, and the sub-network $\psi_k^\star$ with the lowest total rank is selected for the next training stage. Appendix A.4 shows the Algorithm of our EntPruner.

# 3 EXPERIMENT

## 3.1 EXPERIMENTAL SETUP

We evaluate the effectiveness of our proposed method on DiT-XL/2 and SiT-XL/2 models with an image resolution of $256 \times 256$. We conduct experiments on both diffusion-based DiT and flow-matching-based SiT. All experiments are conducted on a computing platform equipped with 8 NVIDIA A800 GPUs (80 GB), with DiT trained for 240K steps and SiT for 60K steps. We fine-tune SiT and DiT on downstream tasks following the configuration in (Xie et al., 2023; Ma et al., 2024).

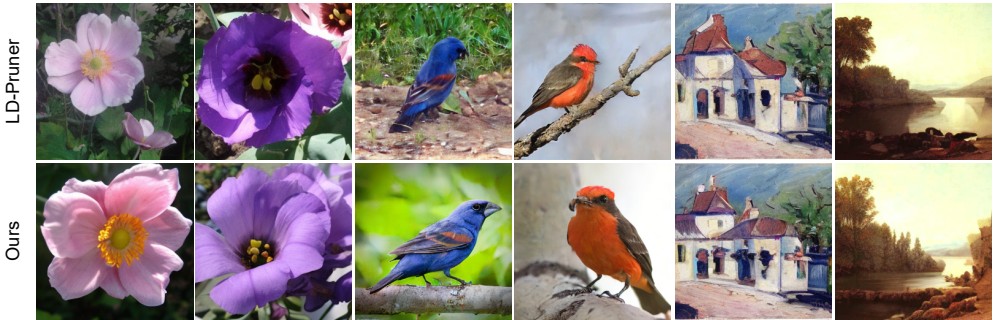

Figure 4: **Qualitative comparison of *flow models* pruned by different methods.** Base model is SiT-XL/2, with 35% pruning rate. Datasets are Flowers (column 1-2), CUB (column 3-4), and ArtBench (column 5-6). Our EntPruner consistently generates finer details and better quality.

Table 1: **Comparison of *flow models* pruned by different methods.** We use SiT as the base model and evaluate with both ODE and SDE samplers on three datasets. ↓ and ↑ indicate whether lower or higher values are better.

| Method | Sparsity | CUB | | Flowers | | ArtBench | | Params (M) | Speedup |
|---|---|---|---|---|---|---|---|---|---|
| | | FID↓ | IS↑ | FID↓ | IS↑ | FID↓ | IS↑ | | |
| Fine-tuning (ODE) | / | 5.32 | 6.02 | 11.78 | 3.71 | 8.80 | 7.32 | 675.12 | ×1 |
| LD-Pruner (ODE) | 35% | 5.70 | 6.03 | 12.02 | 3.75 | 10.78 | 6.63 | 435.78 | ×1.82 |
| **Ours** (ODE) | | **5.48** | **6.07** | **11.75** | **3.82** | **10.03** | **6.88** | **435.78** | **×1.82** |
| LD-Pruner (ODE) | 50% | 6.86 | 6.16 | 12.09 | 3.79 | 12.81 | 6.35 | 334.67 | ×2.22 |
| **Ours** (ODE) | | **6.68** | **6.18** | **11.86** | **3.82** | **12.65** | **6.41** | **334.67** | **×2.22** |
| Fine-tuning (SDE) | / | 5.17 | 5.87 | 12.47 | 3.77 | 13.33 | 6.56 | 675.12 | ×1 |
| LD-Pruner (SDE) | 35% | 5.24 | 6.10 | 12.32 | 3.74 | 16.16 | 6.20 | 435.78 | ×1.49 |
| **Ours** (SDE) | | **5.22** | **6.11** | **12.10** | **3.74** | **15.25** | **6.31** | **435.78** | **×1.49** |
| LD-Pruner (SDE) | 50% | 5.98 | 6.19 | 12.92 | 3.75 | 18.74 | 5.90 | 334.67 | ×1.85 |
| **Ours** (SDE) | | **5.83** | **6.15** | **12.77** | **3.77** | **18.69** | **5.90** | **334.67** | **×1.85** |

Table 2: **Comparison of *diffusion models* trained by different methods.** We use DiT as the base model and compare on three datasets. Our method achieves comparable results with full fine-tuning and other efficient tuning methods with *1.33×* *inference speedup*.

| Method | Sparsity | CUB | Flowers | ArtBench | Params (M) | Speedup |
|---|---|---|---|---|---|---|
| Full Fine-tuning | / | 5.68 | 21.05 | 25.31 | 673.8 | ×1 |
| **Ours** | 30% | **5.50** | **11.99** | **24.99** | **471.66** | **×1.33** |
| Adapt-Parallel | - | 7.73 | 21.24 | 38.43 | 678.08 | - |
| Adapt-Sequential | - | 7.00 | 21.36 | 35.04 | 678.08 | - |
| BitFit | - | 8.81 | 20.31 | 24.53 | 674.41 | - |
| VPT-Deep | - | 17.29 | 25.59 | 40.74 | 676.61 | - |
| LoRA-R8 | - | 56.03 | 164.13 | 80.99 | 674.94 | - |
| LoRA-R16 | - | 58.25 | 161.68 | 80.72 | 675.98 | - |
| DiffFit | - | **5.48** | 20.18 | **20.87** | 674.63 | - |

We select three fine-grained image datasets as the calibration datasets for pruning: CUB-200-2011 (Wah et al., 2011), Oxford Flowers (Nilsback & Zisserman, 2008), and ArtBench-10 (Liao et al., 2022). Notably, ArtBench-10 exhibits a distribution that significantly differs from ImageNet, allowing us to comprehensively evaluate the generalization performance of EntPruner on out of distribution tasks. For the DiT model, we adopt the DDPM sampler, while for SiT, we employ an ODE solver and SDE solver. We report results based on 50 sampling steps. To assess generative quality, we utilize two widely adopted evaluation metrics: Fréchet Inception Distance (FID) and Inception Score (IS). We measure efficiency using inference latency and the number of parameters.

## 3.2 ENTROPY-GUIDED AUTOMATIC PROGRESSIVE PRUNING ON DOWNSTREAM DATASETS

**Class to Image Generation with Flow Models.** To evaluate the effectiveness of our approach, we compare it with LD-Pruner (Castells et al., 2024) on the SiT architecture. As shown in Table 1, our method consistently outperforms LD-Pruner at both 35% and 50% pruning ratios, when pruning 35% of the parameters, we achieve 3.9%, 2.2%, and 6.9% FID improvements on the three benchmark datasets with an ODE sampler. Notably, on Flowers dataset, our method surpasses full fine-tuning when pruning 35% of the parameters, achieving the best overall performance. We observe slight variations across different samplers, with the SDE sampler exhibiting longer sampling times compared to the ODE. Nevertheless, our method consistently outperforms LD-Pruner regardless of the sampling strategy. While both pruning methods exhibit performance degradation on ArtBench, likely due to the substantial distribution gap between ArtBench and ImageNet, however, EntPruner still outperforms LD-Pruner, demonstrating strong robustness and superior generalization.

We qualitatively compare the image generation quality of EntPruner and LD-Pruner. In these experiments, we use the same random seed, set the classifier-free guidance scale to 4.0, and adopt 250 sampling steps. As shown in Figure 4, our method consistently produces images with finer details and better quality. On the CUB and Oxford Flowers datasets, EntPruner tends to generate close-up views of the subject, enriching detail and enhancing aesthetic quality. On the ArtBench dataset,

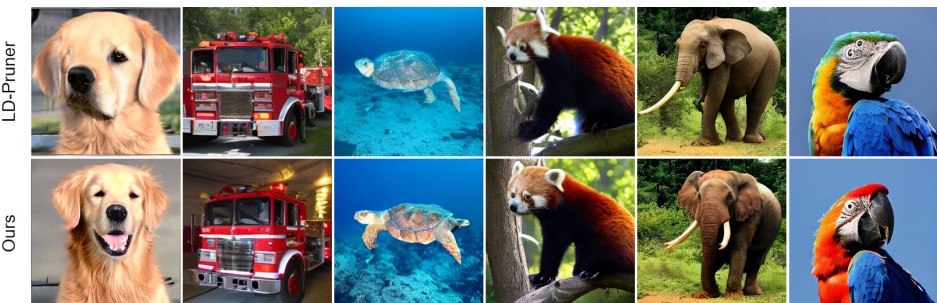

Figure 5: **Qualitative comparison of *flow models* pruned by different methods on ImageNet 256×256.** Base model is SiT with 30% pruning rate. Our EntPruner consistently produces more photorealistic images with *fewer generation artifacts*.

our approach captures more realistic textures in structures such as houses and trees, resulting in improved visual fidelity compared to LD-Pruner.

**Class to Image Generation with Diffusion Models.** We apply EntPruner on DiT and compare with full fine-tuning. As shown in Table 2, our method outperforms full fine-tuning across all three datasets. Notably, on Flowers, EntPruner reduces the FID relatively by **43.04%**. Furthermore, EntPruner improves inference efficiency, achieving a **1.33×** speedup compared to full fine-tuning. In addition, we benchmark EntPruner against *state-of-the-art* fine-tuning strategies. Our method matches or surpasses the performance of efficient fine-tuning baselines, while offering **1.33×** faster inference.

**Inference Efficiency Across Parameter Budgets.** Figure 6 compares the FID scores *vs.* computational complexity (measured by MACs) of EntPruner and LD-Pruner across three datasets: CUB, Flowers, and Art-Bench. Notably, EntPruner consistently outperforms LD-Pruner across all configurations. At medium complexity, the gap between the two is most obvious. Table 1 and Figure 6 jointly demonstrate that EntPruner maintains lower FID scores even under reduced MACs on

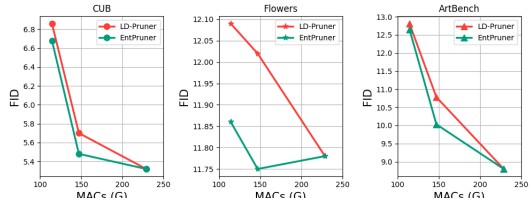

Figure 6: **Tradeoff between computational cost (MACs) and generative quality (FID).** The rightmost point represents the full model.

CUB and Flowers datasets, demonstrating better robustness in resource-constrained settings. On ArtBench dataset, the performance gap between the two methods is most pronounced, further highlighting EntPruner's superior generalization ability and pruning efficiency for complex image generation tasks. See Appendix A.2 for more details.

## 3.3 ENTROPY-GUIDED AUTOMATIC PROGRESSIVE PRUNING ON IMAGENET 256×256

To further demonstrate the effectiveness and robustness of our pruning method on pretrained models, we directly apply our pruning strategy to compress models pretrained on ImageNet 256×256. The experimental setup follows the same configuration as Section 3.1, where SiT is trained using flow matching. During sampling, we employ an ODE solver and adhere to standard evaluation protocols (Peebles & Xie, 2023; Ma et al., 2024).

As shown in Table 3, with a pruning ratio of 30% on ImageNet, our method achieves a final FID of **3.53**, only a degradation of 1.38 compared to the original SiT-XL/2. Notably, it surpasses the recent pruning method LD-Pruner by 48.16%, further validating our

Table 3: **Comparison of *flow models* pruned by different methods on ImageNet 256×256.** Base model is SiT with 30% pruning rate. We also include full model performance of different generative models as a reference.

| Method | Params(M) | FID↓ | Speedup |
|---|---|---|---|
| BigGAN-deep | 112 | 6.95 | - |
| StyleGAN-XL | 166 | 2.30 | - |
| ADM | 554 | 10.94 | - |
| LDM-4-G | 400 | 3.60 | - |
| DiT-XL/2 (DDPM) | 675.12 | 2.27 | ×0.43 |
| SiT-XL/2 (ODE) | 675.12 | 2.15 | ×1 |
| LD-Pruner (SiT,ODE) | 471.66 | 6.81 | ×1.33 |
| Ours (SiT,ODE) | **471.66** | **3.53** | **×1.33** |

method's ability to mitigate parameter collapse often caused by one-shot pruning. Moreover, in terms of inference speed, our method achieves a 1.33× speedup compared to SiT, and a **209.30%** speed improvement compared to DiT, highlighting its practical inference efficiency.

As shown in Figure 5, we present qualitative results demonstrating the impact of pruning on image generation quality. It is evident that pruning with LD-Pruner leads to significant degradation in visual fidelity. For instance, generated images exhibit missing features such as the eyes of a dog, fine details of a raccoon, aesthetically pleasing text on hot air balloons, well-structured cakes, and the tusks of an elephant. In contrast, our method preserves more structural details and visual coherence. By pruning at more appropriate stages, our approach enables the final model to maintain robust generative performance while retaining essential semantic content.

## 3.4 ABLATION STUDY

**Entropy-Guided Pruning.** We perform an ablation study to evaluate the effectiveness of entropy-guided importance ranking. Random pruning leads to high variance and instability, so we prune blocks with the highest transfer entropy, which are identified as most critical by our metric. As illustrated in Table 4, pruning blocks with high entropy causes a substantial performance drop, which cannot be fully recovered even through continued fine-tuning. which proves that transfer entropy is a reliable indicator of parameter importance in pretrained networks.

Table 4: **Ablation studies on Auto pruning and Entropy-Guided pruning.** Results are reported on SiT using Oxford Flowers.

| Method | FID ↓ | IS ↑ | Params | Speedup |
|---|---|---|---|---|
| Full Fine-tuning | 11.78 | 3.71 | 675.12 | ×1 |
| w/o Entropy-Guided | 12.06 | 3.81 | 435.78 | ×1.82 |
| w/o Auto Pruning | 11.84 | 3.80 | 435.78 | ×1.82 |
| Ours | **11.75** | **3.82** | **435.78** | **×1.82** |

**Automated Pruning.** We conduct an ablation study on automated pruning. The results are presented in Table 4, demonstrating that one-shot pruning or premature pruning significantly degrades model performance. In contrast, our method autonomously determines both the timing and extent of pruning, allowing the model to maintain competitive performance, while achieving lower FID scores, and even converging to better solutions than full fine-tuning in some cases.

## 4 RELATED WORK

**Diffusion Models** (Ho et al., 2020; Zhang et al., 2023; Ruiz et al., 2023) have achieved remarkable progress in image synthesis, with recent designs shifting from U-Nets to Transformer backbones for better scalability (Peebles & Xie, 2023; Ma et al., 2024; Chen et al., 2024). While these works focus on boosting generative quality, our approach instead addresses their efficient deployment via entropy-guided pruning.

**Automated Machine Learning.** AutoML automates model design and optimization (Liu et al., 2018; Tan et al., 2019; Cubuk et al., 2019). NAS approaches fall into multi-shot (Real et al., 2019), one-shot (Li et al., 2020), and zero-shot (Lin et al., 2021; Yang & Liu, 2024; Li et al., 2024b) categories. Unlike traditional NAS that searches from scratch, we leverage zero-shot proxies to prune pretrained diffusion Transformers, combining AutoML efficiency with pretrained stability.

**Efficient Inference for Diffusion Models.** The inference cost of diffusion models is primarily influenced by the number of inference steps and the computational cost. The former includes advanced solvers (Lu et al., 2022) and distillation (Lin et al., 2024a; Ren et al., 2024), while the latter explores pruning (Castells et al., 2024; Fang et al., 2023), quantization (Li et al., 2023b), and adaptive compression (Lu et al., 2023; Chen et al., 2023a; Guo et al., 2020). Our work extends this line by reframing block-level pruning as a zero-shot NAS problem, achieving lightweight yet performant diffusion Transformers.

## 5 CONCLUSION AND LIMITATION

We present an entropy-guided automatic pruning framework that leverages zero-shot NAS to adaptively rank block importance and determine pruning ratios across tasks. Our method preserves pretrained knowledge, mitigates performance collapse, and achieves faster inference while maintaining generation quality comparable to full fine-tuning. A potential limitation is that its ease of deployment may also enable misuse in unregulated or adversarial scenarios.

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

# A  APPENDIX

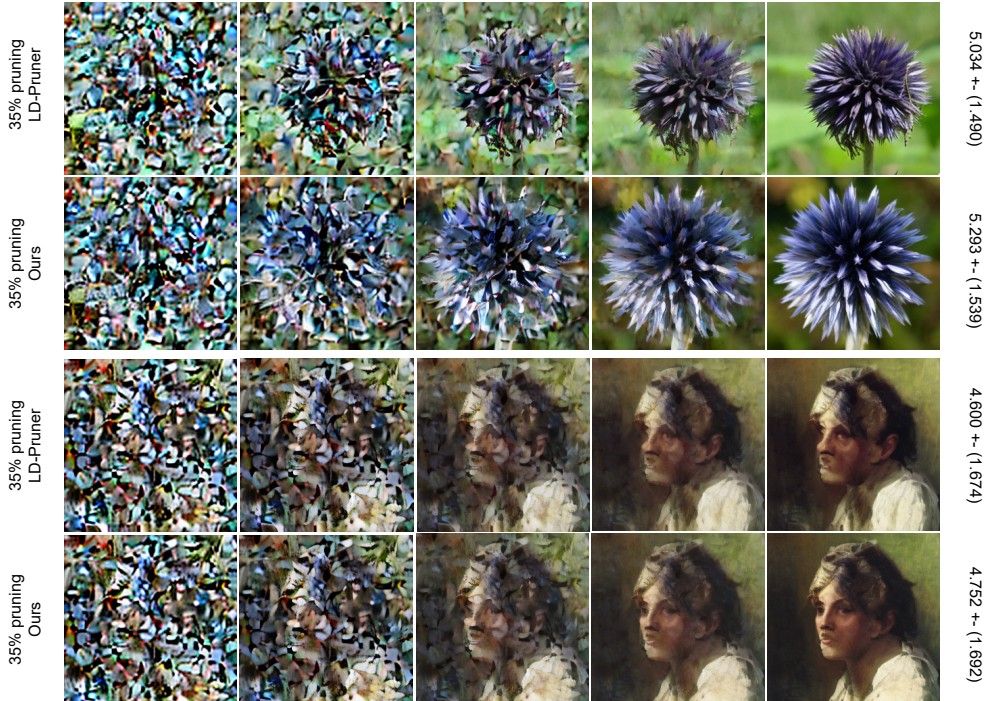

Figure 7: **Visualization of the sampling process of SiT models pruned with different methods.** From left to right, each column corresponds to sampling steps 50, 100, 150, 200 and 250, respectively.

## A.1  PERFORMANCE OF DIFFERENT METHODS

To better understand the generative behavior of different methods, we visualize the denoising trajectories starting from the same noise latent. As shown in Figure 7, we track the intermediate outputs along the denoising path for each method. While both models begin from identical noise, our method consistently produces more visually appealing and coherent results compared to LD-Pruner. In addition, we evaluate the aesthetic quality of generated images using the Neural Image Assessment (NIMA) metric (Talebi & Milanfar, 2018). NIMA employs a trained deep convolutional neural network to predict how users would rate an image in terms of technical quality and aesthetic appeal. Experimental results show that our method

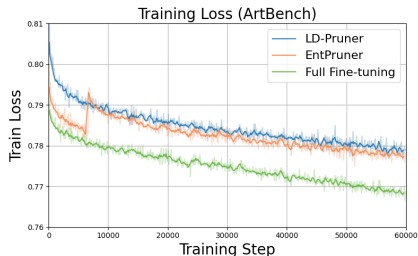

Figure 8: Comparison of training loss trajectories across different methods.

outperforms LD-Pruner by 0.259 and 0.152 points in aesthetic scores, respectively, demonstrating the effectiveness of our approach in preserving visual quality. Figure 8 illustrates the loss trajectory during training with SiT on ArtBench dataset. Owing to our automated pruning strategy, model compression is performed progressively, avoiding abrupt parameter collapse. This leads to faster and more stable convergence.

## A.2  INFERENCE EFFICIENCY ACROSS PARAMETER BUDGETS.

We evaluate the inference efficiency of our method applied to both SiT and DiT under varying parameter scales. FID is reported as the average across three benchmark datasets. Multiply–Accumulate Operations (MACs) are used to quantify the computational complexity of each

Table 5: **Inference Efficiency Evaluation.** We apply our proposed method to both SiT and DiT models to systematically assess inference efficiency under varying parameter budgets.

| Method | Params | Avarage FID | MACs (G) | Latency (s/per) | GPU Memory (GB/per) |
|---|---|---|---|---|---|
| DiT/DDPM | 675.12 | 17.35 | 228.85 | 0.41 | 5.35 |
| | 483.88 | 14.16 | 163.48 | 0.31 | 4.56 |
| SiT/ODE | 675.12 | 8.63 | 228.85 | 0.20 | 5.32 |
| | 435.78 | 9.08 | 147.13 | 0.11 | 4.39 |
| | 334.67 | 10.39 | 114.45 | 0.09 | 3.99 |
| SiT/SDE | 675.12 | 10.32 | 228.85 | 0.76 | 5.36 |
| | 435.78 | 10.85 | 147.13 | 0.51 | 4.38 |
| | 334.67 | 12.43 | 114.45 | 0.41 | 3.93 |

model. In addition, per-image inference time is measured with a batch size of 256. Peak memory usage is recorded as maximum GPU memory required for a single-image inference on A800.

As shown in Table 5, on DiT, pruning 30% of parameters leads to an 18.4% improvement in average FID compared to full fine-tuning. This result reinforces the insight that large-scale models often contain substantial parameter redundancy when transferred to downstream tasks—redundancy that can hinder rather than help model performance. Our pruning strategy is block-level, which allows the computational complexity (as measured by MACs) to decrease proportionally with reduction in parameter count. For instance, when the parameter count is reduced by 35% and 50%, the corresponding MACs are also reduced by approximately 35% and 50%. We further observe that for SiT, using the Euler-based SDE sampler results in the slowest inference speed, whereas the ODE sampler offers the fastest. At a pruning rate of 35%, the FID scores of both samplers decrease (by 0.53 and 0.45), while the memory usage decreases by 18.3% and 19.8%.

### A.3 TRAINABILITY VIA THE CONDITION NUMBER OF NTK IN FLOW MATCHING.

The trainability of a neural network reflects its ability to be effectively optimized via gradient descent. While a network with more parameters theoretically possesses greater expressivity, this does not guarantee practical trainability. The Neural Tangent Kernel (NTK) provides a powerful analytical tool for assessing the convergence behavior of deep networks under gradient-based optimization, particularly in infinite-width regime (Jacot et al., 2018; Novak et al., 2022).

During progressive pruning, we evaluate the trainability of a candidate sub-network with parameters $\omega \in \Lambda_k$, inherited from the previous training stage. Let $\mathcal{L}$ be the velocity prediction loss. Using the chain rule, the parameter update $\Delta\omega$ and the corresponding change in predicted velocity $\Delta v$ can be expressed as:

$$\Delta\boldsymbol{\omega} = -\eta\nabla_{\boldsymbol{\omega}}v(\boldsymbol{x}_t)^{\mathsf{T}}\nabla v(\boldsymbol{x}_t)\mathcal{L},$$
$$\Delta v(\boldsymbol{x}_t) = \nabla\boldsymbol{\omega}v(\boldsymbol{x}_t)\Delta\boldsymbol{\omega}$$
$$= -\eta\hat{\Theta}(\boldsymbol{x}_t,\boldsymbol{x}_t)\nabla v(\boldsymbol{x}_t)\mathcal{L}, \tag{11}$$

where $\eta$ is the learning rate, and $\hat{\Theta}(\boldsymbol{x}_t,\boldsymbol{x}_t) = \nabla_{\boldsymbol{\omega}}v(\boldsymbol{x}_t)^{\mathsf{T}}\nabla_{\boldsymbol{\omega}}v(\boldsymbol{x}_t)$ represents the NTK of the velocity prediction network.

In the infinite-width limit, the NTK remains constant throughout training, and the expected output $\mu(x_t)$ evolves as Jacot et al. (2018):

$$\mu(x_t) = (\mathbf{I} - e^{-\eta\hat{\Theta}s})v_t, \tag{12}$$

where $s$ is the training step index. This expression can be diagonalized in the eigenspace of $\hat{\Theta}$, yielding:

$$\mu(x_t)_i = (1 - e^{-\eta\lambda_i s})v_{t,i}, \tag{13}$$

where $\lambda_i$ denotes the $i^{\text{th}}$ eigenvalue of the NTK.

By ordering the eigenvalues as $\lambda_0 \geq \cdots \geq \lambda_m$, it is known that the maximum stable learning rate scales as $\eta \sim 2/\lambda_0$. Consequently, the convergence rate of the slowest mode is governed by $1/\kappa$,

where $\kappa = \lambda_0/\lambda_m$ is the condition number of the kernel. A smaller $\kappa$ indicates better trainability and faster convergence.

Thus, we adopt the NTK condition number as one of our zero-shot NAS metrics:

$$\mathcal{H}\_\kappa(\psi) = \frac{\lambda_0}{\lambda_m}, \tag{14}$$

where $\lambda_0$ and $\lambda_m$ denote the largest and smallest eigenvalues of the NTK, respectively.

### A.4 Algorithm

The algorithm of Entropy-Guided Automatic Pruning is as follows:

---

**Algorithm 1** Entropy-Guided Automatic Pruning

---

**Input**:
$\Gamma$: pruning operator;
$S$: total training iterations;
$s$: training iterations per stage;
$\omega$: pretrained model parameters;
$N$: number of blocks;
$N_s$: pruning stage interval.
**Output**:
Pruned model parameters $\psi$.
**for** $i \in [1, N]$ **do**
    Compute the transfer entropy $\text{TE}_i$ of block $i$ using Eq. (4).

**end**
Rank blocks in ascending order of $\text{TE}_i$.

**for** $t \in [1, S]$ **do**
    **if** $t \bmod N_s = 0$ **then**
        Evaluate zero-shot performance for each candidate sub-network.
        Select the optimal sub-network $\psi_*$ using Eq. (10).
        Apply pruning: $\psi \leftarrow \psi_*$.

    **end**
    Train the pruned network $\psi$ for $s$ iterations.

**end**

---

### A.5 More Qualitative Results

The sampling results produced by applying our method to the DiT model are shown in Figure 9.

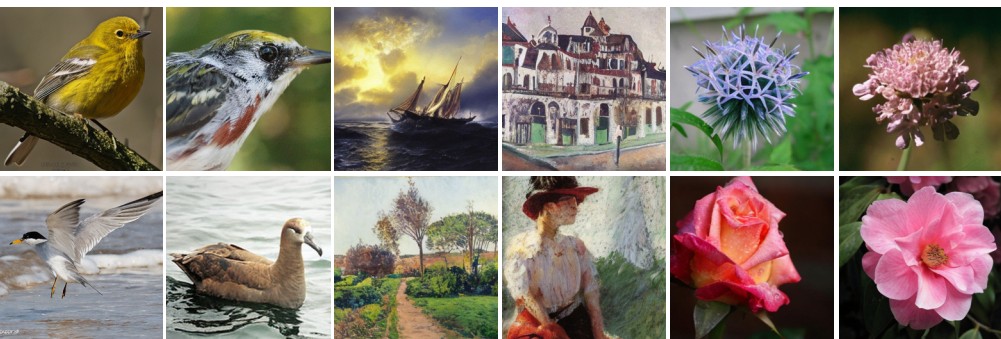

Figure 9: Generated results of a series of *diffusion models* pruned by EntPruner. Base model is DiT-XL/2. The pruning rates are set to 30%. Inference is performed on the CUB (column 1-2), ArtBench (columns 3-4), and Flowers (columns 5-6) datasets, with the classifier-free guidance coefficient set to 4.0. The sampling process involves 250 steps.

