# OpenReview forum: "Entropy-Guided Automated Progressive Pruning for Diffusion and Flow Models"
_ICLR.cc/2026/Conference — ICLR 2026 Conference Withdrawn Submission_

### Official Review · Reviewer_7ePD · 2025-10-30

**Soundness:** 2
**Presentation:** 3
**Contribution:** 2
**Rating:** 4
**Confidence:** 4

**Summary:**

The paper proposes an automated pruning framework for diffusion / flow-based generative models that aims to reduce computation while preserving image quality. The approach has two key components. First, it estimates the importance of each block using an entropy-based criterion: the authors approximate the model’s intermediate output distribution as Gaussian, compute its entropy, and then define a “transfer entropy” score as the entropy reduction attributable to a given block by masking that block during forward passes. Second, instead of using hand-tuned pruning schedules, the method couples this importance ranking with a zero-shot neural architecture search style selector based on NTK conditioning and ZiCo-like gradient proxies to decide how aggressively to prune at each stage. The complete pipeline progressively removes channels/heads/layers and then finetunes the surviving subnet. Experiments on image generation benchmarks (e.g., CUB, Flowers, ArtBench) with SiT/DiT-style architectures claim up to ~2× speedup with limited FID/IS degradation. Overall the paper positions itself as a step toward automated, importance-aware compression for modern diffusion-like generators.

**Strengths:**

The work tackles an important and timely problem: diffusion and flow models are notoriously expensive at inference time, and structured pruning is an appealing alternative to distillation-style acceleration. The paper’s attempt to unify (i) a per-block “information contribution” metric and (ii) an automated pruning schedule/search mechanism is conceptually neat, and goes beyond simple magnitude pruning or uniformly dropping channels. The notion of measuring how much uncertainty a block removes from the model’s own predictions is intuitively aligned with the role of intermediate denoising/score blocks in diffusion. The staged pruning procedure, coupled with an automated search over candidate subnets using zero-shot proxies, is also attractive from an engineering standpoint: in principle it could reduce manual hyperparameter tuning and adapt pruning depth to each model/task pair. Finally, the empirical section covers multiple datasets and reports both quality metrics (FID/IS) and nominal wall-clock improvements, suggesting practical relevance.

**Weaknesses:**

My main concern is that the technical foundations and empirical evidence are not yet strong enough to justify acceptance in the current form. First, the definition of “transfer entropy” used here is nonstandard and seems only loosely related to the classical information-theoretic notion of directed information flow. In the paper, the “transfer entropy” of a block is essentially computed as the difference between (i) the entropy of the model’s intermediate output distribution under normal inference and (ii) the entropy when that block is masked. This is closer to an ablation-style sensitivity score than to true transfer entropy; the manuscript currently blurs that distinction and does not provide a rigorous argument that this difference in (Gaussian-approximated) entropy meaningfully tracks causal importance for generation quality. This conceptual gap weakens the theoretical positioning of the method and makes it hard to assess whether the metric is principled or just an empirical heuristic.

Second, even if we accept entropy reduction as an importance surrogate, the way entropy is estimated is quite aggressive. The method appears to approximate the high-dimensional intermediate feature distribution as a (factorized or effectively scalar-variance) Gaussian, so that entropy boils down to something like (\log \sigma). This collapses spatial/channel correlations and any multimodality of internal activations. In diffusion models, intermediate features can be highly structured, timestep-dependent, and in some cases multimodal across noise seeds. Reducing that to a single scalar variance per block may be too coarse. The paper does not study how sensitive the block ranking is to this Gaussian/variance approximation versus richer statistics (e.g., low-rank covariance, per-channel covariance). Without such an analysis, it is difficult to know whether the pruning policy is stable, or whether it is in fact almost equivalent to a simple variance/magnitude heuristic.

Third, the block masking procedure used to measure entropy-conditioned importance is not obviously equivalent to actually pruning that block in the final network. Masking a block during a forward pass can perturb normalization layers, residual connections, and downstream statistics in a way that is different from structurally removing the block and then finetuning. The paper informally argues that blocks with low “transfer entropy” can be pruned with minimal harm, but I did not see a systematic study of correlation between the proposed score and the eventual FID/IS drop after true pruning and short finetuning. In other words, we need curves like: rank blocks by the proposed score, prune the worst-k%, finetune for X steps, and report final quality vs. the same experiment using random pruning, magnitude-based pruning, or gradient-sensitivity pruning. Without this ablation, it is hard to conclude that the entropy score is actually better than standard pruning heuristics.

Fourth, the second stage of the pipeline leans heavily on zero-shot NAS-style proxies (NTK conditioning, ZiCo-like criteria) to automatically choose pruning stages and ratios. These proxies have been explored primarily on classification networks and supervised tasks, and they are known to sometimes have unstable or task-dependent correlation with final accuracy. Here they are assumed to generalize to generative diffusion backbones, but the paper does not report any empirical rank-correlation analysis between these proxy scores and final image quality metrics such as FID/IS. In the absence of such validation, it is not yet convincing that the automated schedule/search truly “knows” which subnetworks are better, as opposed to performing a heuristic search that happened to work on the reported models.

Fifth, the experimental comparison set feels too narrow for a pruning/compression paper in 2025. The diffusion acceleration literature is broad, including: structured pruning methods tailored to diffusion U-Nets and Transformers (e.g., LD-Pruner and related layer/channel dropping approaches), progressive distillation and consistency-model-based acceleration (which reduce sampling steps and thus wall-clock time), and various low-rank / factorization / token dropping baselines. The paper references some of these families but does not always reproduce them under a unified protocol, same hardware, same batch size, and same privacy/compute budget. As a result, it is difficult to interpret claims such as “2.2× speedup with comparable FID”: are we comparing to the strongest known pruning baseline, to consistency/distillation-style accelerators, or just to the uncompressed teacher? Especially for a paper that advertises practical acceleration, the absence of a single, apples-to-apples end-to-end latency and throughput table (same GPU/CPU backend, same scheduler, same sampler steps) across all baselines is a noticeable gap.

Finally, there is an important scalability / overhead question that the paper largely sidesteps. To compute the entropy-based importance, the method must (a) run forward passes with each block masked to estimate conditional entropy and (b) gather statistics across noise seeds / timesteps. For modern diffusion transformers with dozens of blocks, this is potentially expensive. That overhead should be counted against the claimed acceleration, particularly if the pruning pipeline is meant to be automated per task/domain. Right now the paper frames the method as “automated,” but does not quantify how costly that automation itself is in wall-clock terms relative to, say, straightforward knowledge distillation. This missing accounting weakens the practicality claim.

**Questions:**

(1) How robust is the proposed “transfer entropy” score as a predictor of final generative quality after actual pruning (not masking) and short finetuning? Concretely, can the authors provide rank-correlation (e.g., Spearman) between block importance scores and the FID/IS degradation observed when pruning those blocks, and contrast this with simpler alternatives such as magnitude-based pruning, random pruning, or gradient-sensitivity pruning?

(2) The entropy estimation assumes (approximately) Gaussian feature statistics so that entropy essentially depends on a scalar variance term. Why is this modeling choice justified for high-dimensional, timestep-dependent diffusion features, which are typically structured and sometimes multimodal across noise seeds? Have the authors tried richer covariance models, or at least channel-wise covariance, and if so does the block ranking (and pruning outcome) change materially?

(3) The automated stage selection / pruning schedule relies on zero-shot NAS proxies (NTK conditioning, ZiCo-like criteria) that were originally validated on supervised classification. What evidence do we have that these proxies correlate with downstream generative metrics in this setting? Can the authors report an experiment where several candidate subnetworks are ranked by these proxies and then actually finetuned/evaluated to show quantitative correlation with FID/IS?

(4) The experimental section reports up to ~2× speedup, but the comparison appears to be mostly against the original unpruned model. For a fair assessment of practical acceleration, can the authors present a unified table with end-to-end latency and throughput (same hardware, batch size, sampler/scheduler backend) alongside strong baselines such as LD-Pruner–style structured pruning, progressive/consistency distillation that reduces sampling steps, and low-rank / token-dropping approaches? Without that head-to-head comparison, it is difficult to judge whether the proposed method is actually competitive as a deployment strategy.

Overall, while the paper is motivated by an important problem and contains interesting ideas, these conceptual and empirical gaps make it difficult for me to be confident in the generality and reliability of the approach at this stage.

---

### Official Review · Reviewer_sZeC · 2025-10-30

**Soundness:** 3
**Presentation:** 3
**Contribution:** 3
**Rating:** 6
**Confidence:** 5

**Summary:**

This paper proposes EntPruner, an entropy-guided automatic progressive pruning framework for transformer-based diffusion and flow models. By leveraging transfer entropy to rank block importance and employing zero-shot Neural Architecture Search (NAS), EntPruner dynamically determines pruning schedules to reduce model size while preserving performance. Experiments demonstrate up to 2.22× inference speedup with minimal degradation in generative quality across multiple datasets. The method significantly improves efficiency and generalization, offering a scalable solution for deploying large generative models in resource-constrained settings.

**Strengths:**

+ The method demonstrates strong performance on downstream tasks.
+ The proposed block pruning approach achieves a high acceleration ratio.

**Weaknesses:**

The paper overall is good. But I still have some questions/suggestions.

1. The proposed method appears to be evaluated only on downstream tasks, despite being positioned as a general approach. It should ideally be applicable not only to downstream tasks but also to upstream datasets such as ImageNet. While I understand that fine-tuning on ImageNet can be costly, I strongly recommend the authors make an effort to include such an experiment. This would significantly broaden the scope and impact of the paper.

2. Figure 3 does not clearly distinguish between the two stages. It should be improved by explicitly labeling them as "Ranking Stage" and "Pruning Stage."

3. It would be beneficial to cite and discuss dynamic models such as DyDiT [1], which also includes experiments on transferring to downstream tasks.

4. Experiments under fewer steps (e.g. 50 and 20) should be added.


[1] Dynamic Diffusion Transformer, ICLR 2025




Minor:
1. When "Transfer Entropy" first appears in the Introduction, the authors do not cite any references, making it difficult to understand its purpose and functionality.

**Questions:**

see weakness.

---

### Official Review · Reviewer_uex6 · 2025-11-03

**Soundness:** 2
**Presentation:** 1
**Contribution:** 2
**Rating:** 4
**Confidence:** 3

**Summary:**

The authors try to propose a method for pruning the transformer-based diffusion and flow model. The primary contribution includes the use of the entropy to measure the information of a masked block; the introduction of the condition number to measure the stability and convergence; and the ratio between the gradients to measure the convergence and generation efficiency. They conduct experiments on ImageNet and achieve a 50% decrease in the computational complexity at the cost of generation performance.

**Strengths:**

The authors give good intuition and justification about the importance of pruning.

The authors manage to get a 50% pruning rate.

**Weaknesses:**

The paper's writing needs to be improved. There are a few typos and inconsistencies. For instance, in equation 1, gamma is not defined, and also not used later.

In Figure 2, the authors said they observed a strong positive correlation between the entropy and the resulting loss of the masked block, but it's not clear from the current results, as the increase of entropy does not seem to be correlated naturally with the loss.

**Questions:**

A 1.76 decrease in FID is actually a lot. Have you compared your method to other models with low computational cost (without pruning) but good generative performance?

How to achieve a balance between the three metrics. Now it looks like a heavy parameter tuning.

---

### Official Review · Reviewer_LhhB · 2025-11-05

**Soundness:** 1
**Presentation:** 2
**Contribution:** 3
**Rating:** 2
**Confidence:** 4

**Summary:**

This paper proposes EntPruner, a pruning framework for transformer-based diffusion and flow models. The method consists of two main components: (1) using "transfer entropy" to rank block importance by measuring H(X_out) - H(X_out | Mask{block_i}), and (2) employing zero-shot NAS metrics (NTK condition number and ZiCo) to automatically determine pruning schedules during training. Experiments on DiT-XL/2 and SiT-XL/2 demonstrate up to 2.22× inference speedup with competitive generation quality on ImageNet and three downstream datasets (CUB, Flowers, ArtBench).

**Strengths:**

1. **Strong empirical results**: Achieves 2.22× speedup at 50% pruning with average FID increase of only 1.76 (Table 1). On ImageNet, 30% pruning yields FID 3.53 vs LD-Pruner's 6.81—a substantial 48.16% improvement (Table 3).
2. **Comprehensive experimental coverage**: Tests multiple architectures (DiT, SiT), samplers (DDPM, ODE, SDE), and datasets including out-of-distribution evaluation (ArtBench). Qualitative results (Figures 4-5) show clear visual quality improvements over LD-Pruner.
3. **Novel application**: To my knowledge, this is the first work applying entropy-based importance metrics to diffusion model pruning. The combination with zero-shot NAS for automatic schedule determination is elegant.
4. **Thorough comparison with efficient fine-tuning**: Table 2 shows favorable comparison against DiffFit, LoRA, and other parameter-efficient methods while providing inference speedup they cannot offer.
5. **Progressive pruning framework**: The staged approach (k=4) with automatic sub-network selection mitigates catastrophic forgetting, as evidenced by more stable training curves (Figure 8).

**Weaknesses:**

1. **Unclear theoretical foundation**: The formulation H(X_out) - H(X_out | Mask{block_i}) in Eq. 4 mathematically corresponds to mutual information I(Block_i; Output), not transfer entropy. Transfer entropy as defined by Schreiber (2000) requires:
    - Temporal ordering: measuring how X's past influences Y's future
    - Conditioning on Y's past: TE_{X→Y} = I(Y_{n+1}; X_n^{(k)} | Y_n^{(k)})

    Your formulation lacks both. Could you clarify: (a) Why is this called "transfer entropy"? (b) What is the theoretical justification for treating architectural depth as causality? (c) Have you compared against simply using mutual information or conditional entropy?

2. **No diffusion-specific contributions**: Despite claiming methods tailored for "diffusion and flow models" (Abstract), the approach does not leverage their unique temporal structure:
    - No timestep-dependent importance analysis (varying importance across t)
    - No noise schedule consideration (α_t, σ_t in Eq. Section 2.1 are never used)
    - No trajectory deviation analysis despite discussing flow matching ODEs
    - Section 2.4's "Flow Matching specific" NTK formulation (Eq. 6) is actually generic gradient descent dynamics

    **Question**: Would your method work identically on non-diffusion transformers (e.g., Vision Transformers for classification)? If yes, the "tailored for transformer-based diffusion and flow models" claims seem overstated.

3. **Missing critical ablations** (Table 4 is insufficient):
    - No comparison: TE vs. mutual information vs. conditional entropy vs. magnitude-based pruning
    - No timestep analysis: Does block importance vary across diffusion timesteps t?
    - No hyperparameter sensitivity: How sensitive to k (number of stages), γ (regularization), D (batches for ZiCo)?
4. **Zero-shot proxy validation missing**: You use NTK condition number and ZiCo to select sub-networks, but never show:
    - Correlation between H_κ/H_ZiCo and actual FID/IS
    - How often the selected sub-network is actually optimal
    - Why equal weighting (Eq. 10) is appropriate vs. learned weights
5. **Limited baseline comparison**: Only LD-Pruner is compared in main results. Diff-Pruning (Fang et al., 2023) and BK-SDM (Kim et al., 2024) are cited but not compared. Why not include magnitude-based or gradient-based pruning as basic baselines?

**Questions:**

1. **On Transfer Entropy**: Can you provide mathematical justification for why Eq. 4 constitutes transfer entropy rather than mutual information? Specifically, where is the temporal causality (past→future) in your formulation?
2. **On Diffusion-Specific Design**: Can you show timestep-dependent importance analysis? For example, do different blocks dominate at different noise levels (large t vs. small t)? This would strongly support your diffusion-specific claims.
3. **On Comparison with MI**: What happens if you replace TE with simple mutual information I(Block_i; Output) computed via kernel density estimation? Is there a meaningful difference?
4. **On Ablations**: Can you provide results comparing TE-based importance vs. magnitude-based (||W||) vs. gradient-based (||∂L/∂W||) pruning? This would clarify the value of the information-theoretic approach.

---

### Note · Authors · 2025-11-14

I have read and agree with the venue's withdrawal policy on behalf of myself and my co-authors.